# *sel*Seq: A method for the enrichment of non-polyadenylated RNAs including enhancer and long non-coding RNAs for sequencing

**Jason D. Limberis**[1]*, **Alina Nalyvayko**[1], **Joel D. Ernst**[1], **John Z. Metcalfe**[2]

**1** Division of Experimental Medicine, University of California, San Francisco, San Francisco, CA, United States of America, **2** Division of Pulmonary and Critical Care Medicine, Zuckerberg San Francisco General Hospital and Trauma Centre, University of California, San Francisco, San Francisco, CA, United States of America

* jason.limberis@ucsf.edu

## Abstract

Non-polyadenylated RNA includes a large subset of crucial regulators of RNA expression and constitutes a substantial portion of the transcriptome, playing essential roles in gene regulation. For example, enhancer RNAs are long non-coding RNAs that perform enhancer-like functions, are bi-directionally transcribed, and usually lack polyA tails. This paper presents a novel method, *sel*Seq, that selectively removes mRNA and pre-mRNA from samples enabling the selective sequencing of crucial regulatory elements, including non-polyadenylated RNAs such as long non-coding RNA, enhancer RNA, and non-canonical mRNA.

## Introduction

Noncoding RNAs (ncRNAs), a large subset of which are non-polyadenylated (polyA), are crucial regulators of RNA expression in a diverse range of life forms [1–3]. In mammals, ncRNAs constitute a substantial portion of the transcriptome [1], playing essential roles in gene regulation via chromatin modification transcriptional regulation, and post-transcriptional processing [4]. Long non-coding RNAs (lncRNAs) are the most prevalent and diverse class of ncRNAs and are typically over 200 nucleotides long and lack protein-coding potential [1]. A large subset of lncRNAs is not polyadenylated and regulate gene expression, with abnormal expression associated with several diseases [5–7] and are predictive of outcomes [8]. Enhancer RNAs (eRNAs) are lncRNAs that perform enhancer-like functions [5, 6], are bi-directionally transcribed, and usually lack polyA tails [9]. Due to their low copy numbers [10, 11], eRNAs are challenging to detect, with most being non-polyA. Nevertheless, measuring eRNAs using total RNA-seq has provided valuable insights into transcriptional regulation [12].

Current methods to explore ncRNAs and intermediate messenger RNA splicing reactions, such as total RNA sequencing or microarrays, have severe limitations. For example, microarrays require knowledge of the transcripts of interest a priori and have a small dynamic range. Total RNA sequencing quantifies all RNA in a sample, which, after ribosomal RNA depletion, is dominated by mRNA transcripts and includes a relatively low number of lncRNAs (a typical

**Data Availability Statement:** Sequence data is available on SRA under the accession number PRJNA949687.

**Funding:** JZM, 1R01AI153213, National Institute of Allergy and Infectious Diseases (NIAID), https://www.niaid.nih.gov/, The funders did not and will not have a role in study design, data collection and analysis, decision to publish, or preparation of the manuscript.

**Competing interests:** The authors have declared that no competing interests exist.

human cell has $3\times10^3$-$5\times10^4$ lncRNAs, while there are $3\times10^5$-$1\times10^6$ mRNA transcripts). Thus, quantification of rare lncRNAs and isoforms is challenging. Here, we present a novel method that selectively removes polyadenylated RNA (e.g., mRNA and pre-mRNA; and optionally rRNA) from samples, allowing for the comprehensive characterization of low-abundance lncRNAs, eRNAs, and non-canonical mRNA splicing. Our method, *sel*Seq, offers a reliable and sensitive approach to exploring these crucial regulatory elements.

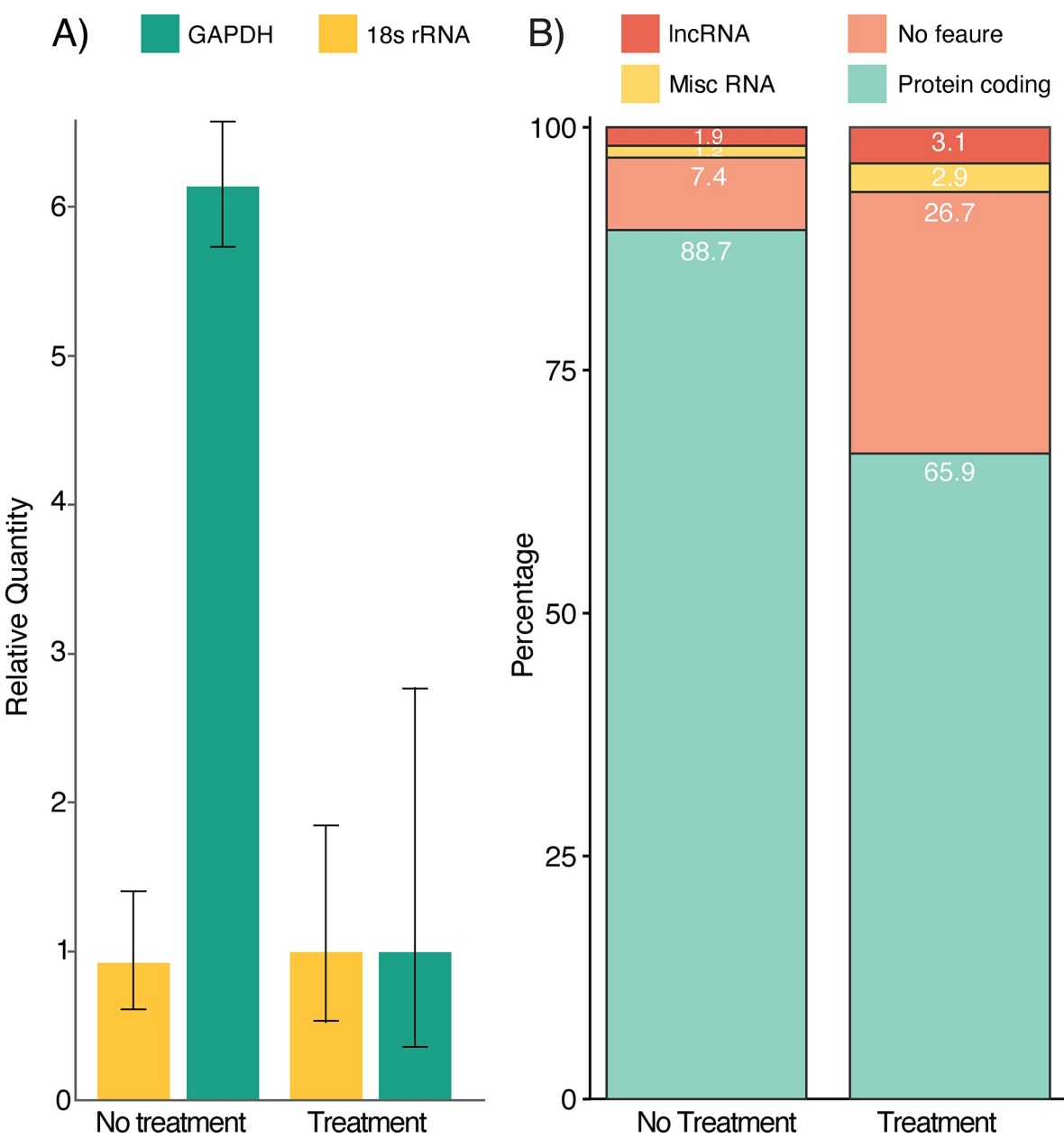

**Fig 1.** *sel*Seq results showing A) the decrease in polyA tailed *GAPDH* housekeeping gene and no decrease in non-polyA tailed 18s ribosomal RNA (n = 4; error bars show one standard deviation); and B) total RNA sequencing after rRNA depletion shows an increase in the proportion of long non-coding, unassigned (no feature), and miscellaneous RNA with a corresponding decrease in the protein-coding assignments, the remainder of which are likely non-polyadenylated transcripts.

## Materials and methods

The protocol described in this peer-reviewed article is published on protocols.io, DOI 10.17504/protocols.io.j8nlkwpk6l5r/v1, and is included for printing as S1 File with this article.

## Expected results

There will be a decrease in the RNA quantity following the completion of the protocol. A reverse transcriptase quantitative PCR can be used to see the reduction of polyA housekeeping genes to non-polyA rRNA (**Fig 1A**) if the optional rRNA depletion step is not done as part of the protocol. Total RNA sequencing after rRNA depletion shows a large decrease in the proportion of polyA-tailed transcripts as illustrated in **Fig 1B**, where human liver total RNA (Ambion, USA) was used and aligned to the GENCODE nucleotide sequence of the Human GRCh38.p13 genome assembly version containing all regions, including reference chromosomes, scaffolds, assembly patches, and haplotypes. The proportion of lncRNA, no feature, and miscellaneous RNA have increased substantially, and the protein-coding assignments have decreased, with those remaining likely non-polyadenylated transcripts or degraded polyadenylated transcripts.

## Supporting information

**S1 File. The protocol in PDF format available from protocols.io is provided as supporting information File 1, with the caption: S1: Step-by-step protocol, also available on protocols. io.** Sequence data is available on SRA under the accession number PRJNA949687.
(PDF)

## Author Contributions

**Conceptualization:** Jason D. Limberis.

**Formal analysis:** Jason D. Limberis.

**Funding acquisition:** John Z. Metcalfe.

**Investigation:** Jason D. Limberis.

**Methodology:** Jason D. Limberis.

**Project administration:** Jason D. Limberis.

**Software:** Jason D. Limberis.

**Supervision:** Joel D. Ernst, John Z. Metcalfe.

**Validation:** Jason D. Limberis.

**Visualization:** Jason D. Limberis.

**Writing – original draft:** Jason D. Limberis, Alina Nalyvayko.

**Writing – review & editing:** Jason D. Limberis, Alina Nalyvayko, Joel D. Ernst, John Z. Metcalfe.

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
