## [Decision Letter · Decision Letter 0]

7 Jun 2023

PONE-D-23-12460sel Seq: A method for the enrichment of non-polyadenylated RNAs including enhancer and long non-coding RNAs for sequencing.PLOS ONE

Dear Dr. Limberis,

Thank you for submitting your manuscript to PLOS ONE. After careful consideration, we feel that it has merit but does not fully meet PLOS ONE’s publication criteria as it currently stands. Therefore, we invite you to submit a revised version of the manuscript that addresses the points raised during the review process. The comments from two reviewers are at the bottom of the letter. Please address all the points discussed from two reviewers.

We look forward to receiving your revised manuscript.

Kind regards,

Xiaoyong Sun

Academic Editor

PLOS ONE

Reviewers' comments:

Reviewer's Responses to Questions

**Comments to the Author**

1. Does the manuscript report a protocol which is of utility to the research community and adds value to the published literature?

Reviewer #1: Yes

Reviewer #2: Yes

2. Has the protocol been described in sufficient detail?

To answer this question, please click the link to protocols.io in the Materials and Methods section of the manuscript (if a link has been provided) or consult the step-by-step protocol in the Supporting Information files.

The step-by-step protocol should contain sufficient detail for another researcher to be able to reproduce all experiments and analyses.

Reviewer #1: Yes

Reviewer #2: Yes

3. Does the protocol describe a validated method?

Reviewer #1: Yes

Reviewer #2: Yes

4. If the manuscript contains new data, have the authors made this data fully available?

Reviewer #1: Yes

Reviewer #2: Yes

**5. Is the article presented in an intelligible fashion and written in standard English?**

Reviewer #1: Yes

Reviewer #2: Yes

6. Review Comments to the Author

Reviewer #1: In this paper, the authors found a new method to enrich RNA without polyA tails to facilitate the sequencing of enhancers and non-coding RNAs，which is a meaningful study. However, the following questions remain:

Major Points:

1、There are currently Dynabeads carrying Oligo dTs on the market, which can also remove the RNA of polyA tail, what are the advantages of your method?

2、In Fig1B, protein RNA accounted for 65.9% after treatment, which is still a high proportion, and I doubt that it can play a huge role in subsequent sequencing work？

3、Can you use several more genes to verify the efficiency of the your method?

Reviewer #2: This manuscript describes a novel method for the enrichment of non-polyadenylated RNAs including enhancer and long noncoding RNAs for sequencing.

Two strengths of this manuscript are:

A. A reverse transcriptase quantitative PCR used to see the reduction of polyA housekeeping genes to non-polyA Rrna, but there are hardly any data available to show how relative representation is calculated. The authors describe the reduction in relative expression of GADPH polyA housekeeper genes to non-polyA rRNA, but no description with in-group (treatment) or with intergroup (no treatment&treatment).

B. Line 56-58 “The proportion of lncRNA, no feature, and miscellaneous RNA have increased substantially, and the protein-coding assignments have decreased, with those remaining likely non-polyadenylated transcripts.” The relative increase times of the proportion of “no feature “is more than that of other types. The author does not explain the potential types of RNA in “no feature” (whether it may be microRNA) and the possible reasons for the large increase times of no feature.

A smattering of suggested corrections:

1. Figure 1B, the misspelling “no feaure”.

2. In the Supporting Information, the misspelling “Optional: Ane-step RT-qPCR quantification” in the last step.

7. PLOS authors have the option to publish the peer review history of their article (what does this mean?). If published, this will include your full peer review and any attached files.

Reviewer #1: No

Reviewer #2: **Yes: **Xinhua Ding

---

## [Author Response · Author response to Decision Letter 0]

10 Jul 2023

PONE-D-23-12460

sel Seq: A method for the enrichment of non-polyadenylated RNAs including enhancer and long non-coding RNAs for sequencing.

PLOS ONE

Dear Dr. Limberis,

Thank you for submitting your manuscript to PLOS ONE. After careful consideration, we feel that it has merit but does not fully meet PLOS ONE’s publication criteria as it currently stands. Therefore, we invite you to submit a revised version of the manuscript that addresses the points raised during the review process.

The comments from two reviewers are at the bottom of the letter. Please address all the points discussed from two reviewers.

We look forward to receiving your revised manuscript.

Kind regards,

Xiaoyong Sun

Academic Editor

PLOS ONE

Reviewers' comments:

Reviewer's Responses to Questions

Comments to the Author

1. Does the manuscript report a protocol which is of utility to the research community and adds value to the published literature?

Reviewer #1: Yes

Reviewer #2: Yes

2. Has the protocol been described in sufficient detail?

To answer this question, please click the link to protocols.io in the Materials and Methods section of the manuscript (if a link has been provided) or consult the step-by-step protocol in the Supporting Information files.

The step-by-step protocol should contain sufficient detail for another researcher to be able to reproduce all experiments and analyses.

Reviewer #1: Yes

Reviewer #2: Yes

3. Does the protocol describe a validated method?

Reviewer #1: Yes

Reviewer #2: Yes

4. If the manuscript contains new data, have the authors made this data fully available?

Reviewer #1: Yes

Reviewer #2: Yes

5. Is the article presented in an intelligible fashion and written in standard English?

Reviewer #1: Yes

Reviewer #2: Yes

We thank the reviewers for their time and insights and thank you for the opportunity to submit a revised protocol. We have carefully considered each comment and made the necessary revisions to strengthen the manuscript. I appreciate your consideration, and we look forward to hearing from you.

6. Review Comments to the Author

Reviewer #1: In this paper, the authors found a new method to enrich RNA without polyA tails to facilitate the sequencing of enhancers and non-coding RNAs，which is a meaningful study. However, the following questions remain:

Major Points:

1、There are currently Dynabeads carrying Oligo dTs on the market, which can also remove the RNA of polyA tail, what are the advantages of your method?

selSeq offers several advantages over Dynabeads Oligo dTs. To remove poly-A-tailed RNA using the Dynabeads Oligo dT, the supernatant of the incubation must be kept and used, this would require an additional concentration and cleanup step. If Dynabeads Oligo dTs are used, then the poly-A tailed RNA is removed; however, ribosomal RNA cannot be depleted in the same step as it can in selSeq (step 4 in protocol “Optional: rRNA depletion”). Lastly, using Dynabeads Oligo dTs would increase the cost per sample.

2、In Fig1B, protein RNA accounted for 65.9% after treatment, which is still a high proportion, and I doubt that it can play a huge role in subsequent sequencing work？

We agree. This is most likely comprised of degraded transcripts that lost their poly-A tails or transcripts that lacked poly A tails. Depending on the application of this protocol, it may be helpful in investigating the RNA degradome.

3、Can you use several more genes to verify the efficiency of the your method?

In Figure 1A, we show the results of a qRT-PCR the decrease in polyA-tailed GAPDH housekeeping gene and no decrease in non-polyA tailed 18s ribosomal RNA as described in Step 17 “Optional: One-step RT-qPCR quantification” of the protocol. This is a test to confirm that the protocol worked as expected before sequencing. Since we then sequence, we can compare all genes between the two groups, and determine the reduction in the relative number of reads. For example, if we look at the relative expression for five housekeeping genes (ACTB, GAPDH, RPLP0, TBP, PGK1) between the treatment and no treatment in this example, we find 446 (IQR 388, 597) fold higher expression in the No Treatment group. This shows that the method effectively removed these highly expressed, poly-A tailed transcripts.

Reviewer #2: This manuscript describes a novel method for the enrichment of non-polyadenylated RNAs including enhancer and long noncoding RNAs for sequencing.

Two strengths of this manuscript are:

A. A reverse transcriptase quantitative PCR used to see the reduction of polyA housekeeping genes to non-polyA Rrna, but there are hardly any data available to show how relative representation is calculated. The authors describe the reduction in relative expression of GADPH polyA housekeeper genes to non-polyA rRNA, but no description with in-group (treatment) or with intergroup (no treatment&treatment).

We compared the amounts of GADPH and 18s rRNA in different groups by using the Treatment group as a reference. In Figure 1A, we set the relative quantity for the Treatment group as 1, and the error bars represent the variation in the measurements. To calculate this relative quantity, we used a standard method called ΔΔEqCq, which involves determining the fold change between the EqCq values. The EqCq values are obtained by taking the average of the Equivalent Cq values (the PCR cycle number at which the sample's reaction curve intersects the threshold line) for technical replicates of a No Treatment sample and the Treatment reference sample. The summary of these results are in the table below.

╔════════════╤══════╤═══════════════╤═════════════╤════════════════╤══════════════════════╤══════╤══════╗

║Sample │Target│Delta EqCq Mean│Delta EqCq SD│Delta Delta EqCq│Relative quantity (RQ)│RQ Min│RQ Max║

╠════════════╪══════╪═══════════════╪═════════════╪════════════════╪══════════════════════╪══════╪══════╣

║Treatment │18s │10.4 │0.9 │0 │1 │0.5 │1.8 ║

╟────────────┼──────┼───────────────┼─────────────┼────────────────┼──────────────────────┼──────┼──────╢

║Treatment │GAPDH │24.2 │1.5 │0 │1 │0.4 │2.8 ║

╟────────────┼──────┼───────────────┼─────────────┼────────────────┼──────────────────────┼──────┼──────╢

║No Treatment│18s │10.5 │0.6 │0.1 │0.9 │0.6 │1.4 ║

╟────────────┼──────┼───────────────┼─────────────┼────────────────┼──────────────────────┼──────┼──────╢

║No Treatment│GAPDH │21.6 │0.1 │-2.6 │6.1 │5.7 │6.6 ║

╚════════════╧══════╧═══════════════╧═════════════╧════════════════╧══════════════════════╧══════╧══════╝

B. Line 56-58 “The proportion of lncRNA, no feature, and miscellaneous RNA have increased substantially, and the protein-coding assignments have decreased, with those remaining likely non-polyadenylated transcripts.” The relative increase times of the proportion of “no feature “is more than that of other types. The author does not explain the potential types of RNA in “no feature” (whether it may be microRNA) and the possible reasons for the large increase times of no feature.

We used the GENOCDE Human GRCh38.p13 genome assembly containing all regions, including reference chromosomes, scaffolds, assembly patches, and haplotypes and the “comprehensive gene annotation”. The reference annotations we used contain many biotypes, including "protein_coding", "lncRNA", "retained_intron", "pseudogenes", "misc_RNA", and "miRNA". Any sequence that aligns to the reference but contains no information in the annotation file is classified as “no feature”. Since the annotations of non-coding RNAs are not exhaustive, as methods to detect and classify them are few, we think that most of the increase in this group is due to unannotated non-coding RNAs. 

A smattering of suggested corrections:

1. Figure 1B, the misspelling “no feaure”.

2. In the Supporting Information, the misspelling “Optional: Ane-step RT-qPCR quantification” in the last step.

Thanks, we have corrected these.

7. PLOS authors have the option to publish the peer review history of their article (what does this mean?). If published, this will include your full peer review and any attached files.

Do you want your identity to be public for this peer review? For information about this choice, including consent withdrawal, please see our Privacy Policy.

Reviewer #1: No

Reviewer #2: Yes: Xinhua Ding

---

## [Decision Letter · Decision Letter 1]

19 Jul 2023

sel Seq: A method for the enrichment of non-polyadenylated RNAs including enhancer and long non-coding RNAs for sequencing.

PONE-D-23-12460R1

Dear Dr. Limberis,

We’re pleased to inform you that your manuscript has been judged scientifically suitable for publication and will be formally accepted for publication once it meets all outstanding technical requirements.

Kind regards,

Xiaoyong Sun

Academic Editor

PLOS ONE

Additional Editor Comments (optional):

Reviewers' comments:

Reviewer's Responses to Questions

**Comments to the Author**

1. Does the manuscript report a protocol which is of utility to the research community and adds value to the published literature?

Reviewer #2: Yes

2. Has the protocol been described in sufficient detail?

To answer this question, please click the link to protocols.io in the Materials and Methods section of the manuscript (if a link has been provided) or consult the step-by-step protocol in the Supporting Information files.

The step-by-step protocol should contain sufficient detail for another researcher to be able to reproduce all experiments and analyses.

Reviewer #2: Yes

3. Does the protocol describe a validated method?

Reviewer #2: Yes

4. If the manuscript contains new data, have the authors made this data fully available?

Reviewer #2: Yes

**5. Is the article presented in an intelligible fashion and written in standard English?**

Reviewer #2: Yes

6. Review Comments to the Author

Reviewer #2: The author did a great job, sel Seq: A method for the enrichment of non-polyadenylated RNAs including enhancer

and long non-coding RNAs for sequencing. It has been modified as requested.

7. PLOS authors have the option to publish the peer review history of their article (what does this mean?). If published, this will include your full peer review and any attached files.

Reviewer #2: No

---

## [Editor Report · Acceptance letter]

25 Jul 2023

PONE-D-23-12460R1 

*sel*Seq: A method for the enrichment of non-polyadenylated RNAs including enhancer and long non-coding RNAs for sequencing. 

Dear Dr. Limberis:

I'm pleased to inform you that your manuscript has been deemed suitable for publication in PLOS ONE. Congratulations! Your manuscript is now with our production department. 

Kind regards, 

on behalf of

Dr. Xiaoyong Sun 

Academic Editor

PLOS ONE